# Decentralized Federated Learning Over Noisy Labels: A Majority Voting Method

## Abstract

Contrary to centralized federated learning (CFL), decentralized federated learning (DFL) allows clients to cooperate in training their local models without relying on a central parameter server. As different clients have varying annotation skills and preferences, noisy labels are inevitable in decentralized data ownership. In centralized learning (CL) and CFL settings, learning from noisy labels has been extensively explored; however, such methods cannot be directly applied in DFL settings due to limited computational resources or privacy requirements. This paper introduces DFLMV *(majority voting based decentralized federated learning)*, a general DFL framework for learning from noisy data without relying on any assumptions about local client noise models while maintaining data privacy for all clients. Specifically, (1) Clients first use traditional DFL to train their local models until they become stable. (2) Clients use each of their neighbors' models to make a prediction of every data point in their training datasets, then correct the labels based on majority voting. (3) Clients further fine-tune their models based on their updated training dataset. A theoretical analysis of DFLMV is also provided. Extensive experiments conducted on MNIST, Fashion-MNIST, CIFA-10, CIFAR-10N, CIFAR-100N, Clothing1M, and ANIMAL-10N validate the effectiveness of our proposed approach at various noise levels and different data settings in mitigating the adverse effects of noisy labels.

## 1 Introduction

Data labeling is an indispensable step in the data preparation for training deep neural networks (DNNs), as it involves assigning meaningful annotations to newly collected data, thereby making the data interpretable by model training. While accurate data labeling is essential to ensure high quality model training (Chen et al., 2020), noisy labels, such as misinterpretations and neglecting data points, are inevitable in the annotation of large volumes of data. This is because the labeling process typically relies on human annotators to perform the tasks, such as object identification in images, emotion tagging in text, or audio transcription (Wang et al., 2022a), but not every data annotator has all the necessary domain-specific knowledge (e.g., the fine-grained CUB-200 requires ornithologists' expertise (Welinder et al., 2010)) and the full carefulness in labeling every data point. In fact, various studies have shown that noisy labeling is a wide-spread commonly-seen issue (or problem) in the data annotation process, affecting almost all large-scale datasets. For example, a study by MIT found that approximately 10% (5 million data points) of the QuickDraw dataset, 5.83% (2,916 data points) of the ImageNet test set, and 5.85% (585 data points) of the CIFAR-100 test set were mislabeled due to annotators' carelessness and limited knowledge (Holt, 2021).

Recent studies have revealed that low-quality noisy labels can adversely affect many aspects of model training, including the trained model's generalizability, robustness, interpretability, and accuracy, eventually resulting in low-quality models (Chen et al., 2020). This negative impact is even exacerbated under a federated learning (FL) setting, where the model is trained in a distributed way over datasets owned by different clients. As different clients have varying annotation skills, knowledge levels, and attention to detail, some clients' datasets have high-quality labels, while others' do not. Such an unbalanced label quality across different local datasets leads to local models of different qualities, and hence undermining the quality of the global model. Therefore, how to minimize the detrimental effects of noisy labels, which may be unintentionally generated by workers due to

their lack of knowledge or carelessness, so as to retain high-quality training over distributed datasets of diverse label qualities remains a critical challenge for practical FL implementation.

Learning with noisy labels has been extensively studied under both centralized learning (CL) and centralized parameter server-based FL (CFL) (McMahan et al., 2017) settings. At a high level, the methods under CL settings can be divided into three different types: (1) loss correction methods (Wang et al., 2019; Englesson & Azizpour, 2024), (2) clean data preselection methods (Chen et al., 2019; Northcutt et al., 2021), and (3) noisy label correction methods (Tanaka et al., 2018). The methods under CFL settings can be classified into two categories: (1) noisy label correction methods (Xu et al., 2022; Zeng et al., 2022), and (2) noisy label filtering methods (Yang et al., 2021; Li et al., 2024b). However, existing methods are fundamentally limited by their dependence on a powerful central server (many/one-to-one), making them incompatible with decentralized federated learning (DFL) (Koloskova et al., 2019), which does not have a central server but instead relies entirely on peer-to-peer communication (many-to-many) among resource-constrained edge devices (e.g.,connected and automated vehicles (CAVs) and unmanned aerial vehicle constellations (UAVs) (Yuan et al., 2024)). More specifically, the un-applicability of existing methods on DFL is due to the following three main reasons: (1) some methods violate the privacy requirements of DFL. For instance, Englesson & Azizpour (2024) requires all clients' data samples must be accessible directly by the server in order for it to learn the noise transition matrix. However, in DFL, each client must keep their data local. (2) other methods, such as those in Xu et al. (2022); Li et al. (2020); Nishi et al. (2021); Northcutt et al. (2021); Zeng et al. (2022), involve intensive computations under a peer-to-peer setting to select clean labels during training, resulting in high synchronization costs and computation overhead when clients conduct model aggregation. (3) methods in Duan et al. (2022); Li et al. (2024b) require a clean supplementary dataset. However, such a clean dataset is nearly impossible to obtain for DFL, as a client cannot infer clean data for other clients.

In this paper, we focus on learning with noisy labels under the DFL framework, as noisy data presents a more acute problem for this framework due to the lack of a centralized entity to orchestrate the noisy label correction and mitigation process. We expect this work to generate an impact on improving the reliability and accuracy of DFL applications in vital domains such as autonomous-driving vehicles, healthcare, and LEO (Low Earth Orbit) satellites (Yuan et al., 2024).

To mitigate noisy labels in DFL, we propose a three-stage label correction algorithm called DFLMV (*Majority voting based decentralized federated learning*). Specifically, in Stage 1, all clients use traditional DFL to train their local models based on their original local datasets. Once their local models' loss values become stable, clients proceed to Stage 2, where each client exchanges model parameters with its online neighbors and uses each neighbor's model to infer a label for each data point in its local training dataset. Among all inferred labels of the same data point, using majority voting, the client picks the most common one and uses it as the updated label of the data. In Stage 3, based on their updated dataset, each client runs extra training epochs to fine-tune its local model obtained from Stage 1. It is also important to note that this paper addresses the commonly seen non-malicious scenario where label errors arise unintentionally due to annotators' lack of knowledge or recklessness. The malicious attack scenarios, whereby workers/clients collude to inject deliberately fabricated false data and labels, is beyond the scope of this study.

The **main contributions** of this work are summarized as follows:

- A novel majority-voting-based DFL method, DFLMV, is proposed to enable high-quality learning over distributed and noisy-labeled data. In contrast to existing methods, DFLMV has the unique benefits of low computation and communication overhead (as analyzed in Section 4.4), preserving local data privacy, and not requiring supplementary clean datasets.

- We establish two key theoretical performance bounds for DFLMV. Firstly, we derive a general upper bound on the generalization error of any DFL algorithm using cross-entropy loss under arbitrary label noise. Secondly, we derive an upper bound on the error rate of majority voting for a multi-class classification problem. Based on these bounds, we rigorously analyze several factors influencing the error rate of our label correction mechanism and prove that DFLMV guarantees a gain over vanilla DFL (i.e., without MV). To make the theoretical proof of the error rate upper bound mathematically tractable, we assume non-colluding neighbors with identical vote distributions in our proof. To evaluate how effective the proposed DFLMV method can perform in real-world environment, we relax the above assumptions and conduct extensive experiments over seven different datasets under

various non-IID data/noise settings. As detailed in Section 5, DFLMV achieves significant accuracy gains, with accuracy increased by up to 23%, particularly in non-IID settings.

- We conduct extensive experiments on three synthetic datasets (MNIST, Fashion-MNIST, and CIFAR-10) across 12 settings, considering combinations of IID/non-IID data, IID/non-IID noisy labels, and three noise models (symmetric, pairflip, and asymmetric). Additionally, we test DFLMV on four real-world noisy datasets (CIFAR-10N, CIFAR-100N, Clothing1M, and ANIMAL-10N) under non-IID conditions. These experiments verify that the proposed DFLMV approach effectively mitigates the detrimental effects of noisy labels and significantly improves the learned model's accuracy.

Note that even though DFLMV is presented in the context of DFL in this paper, it is also easy to see that the method can be extended to CFL with minor changes, as elaborated in Appendix A.

## 2 RELATED WORKS

**Decentralized Federated Learning.** DFL is an emerging FL framework. With DFL, there is no central server for aggregating model parameters. Clients train their models by exchanging their model parameters with each other without divulging any of their local data during the training process. The concept of DFL was first proposed in Lalitha et al. (2018). In recent years, the DFL structure comes in a wide variety of variants, including sequential pointing line structures (Chang et al., 2018; Sheller et al., 2019; 2020), cycle pointing ring structures (Huang et al., 2022; Yuan et al., 2023), fully connected peer (mesh) structures (Assran et al., 2019; Roy et al., 2019; Chen et al., 2022), hybrid structures (Shi et al., 2021; Wang et al., 2022b), etc. The primary assumption behind these studies is that every client's local dataset is noise-free. However, it has been shown that this assumption cannot be held in a practical DFL system because clients have varying annotation skills and personal preferences (Chen et al., 2020). As noisy labels are inevitable in decentralized data ownership, it is imperative to consider the existence of noisy labels and work on developing an appropriate method to deal with these noisy labels effectively.

**Learning with Label Noise.** Incomplete patterns and cognitive errors can cause label noise. Learning with noisy labels has been extensively explored in CL and CFL settings. Generally speaking, there are three categories of CL methods: (1) loss correction methods (Wang et al., 2019; Englesson & Azizpour, 2024; Hendrycks et al., 2018): These methods often assume noisy labels deteriorate from ground-truth labels due to an unknown noise transition matrix T, and these approaches acquiesce to all clients' data participating in model training to learn this matrix T. (2) clean data preselection methods (Chen et al., 2019; Northcutt et al., 2021): These methods need to perform computation-intensive procedures to select clean data with several cross-validation iterations during training. (3) noisy label correction methods (Xiao et al., 2015a; Li et al., 2017; Tanaka et al., 2018; Vahdat, 2017): These approaches typically require an additional clean dataset for detecting and relabeling noisy labels. Methods in CFL settings can be roughly classified into two categories: (1) label correction methods (Xu et al., 2022; Zeng et al., 2022): Most of these methods involve exchanging both model parameters and additional information with the server to train an auxiliary module for future label correction stages, increasing computing power. (2) noisy label filtering methods (Yang et al., 2021; Li et al., 2024b): These approaches typically require a clean supplementary dataset to train the noisy label filter. However, the above methods cannot be directly applied to the DFL framework due to three main reasons. (1) Methods requiring excessive computational power can lead to high synchronization costs during the aggregation of clients' models. This increases communication overhead for each client's local model convergence, making it difficult to achieve efficient convergence. (2) Methods of acquiescing to all client data participating in model training will violate DFL's privacy policy. (3) Some methods require a clean supplementary dataset. However, such an auxiliary clean dataset is hard to obtain, as noisy labels will be unavoidable in decentralized data ownership. Therefore, it is crucial to develop a practical DFL method that can reduce the negative impact of corrupt labels and ensure high-quality training over diverse datasets with noisy labels.

## 3 PRELIMINARIES

Given a DFL system with $|K|$ clients, where each client $k \in K$ possesses a noisy local training dataset $D_k = \{(x_k(i), y_k(i))\}_{i=1}^{|D_k|}$ (with $x$ being the feature and $y$ being the label), our goal is to let each client $k$ construct an improved dataset $\widetilde{D_k}$, containing less noisy data than $D_k$. Following this,

each client aims to find the optimal solution for minimizing the following empirical risk function:

$$\underset{w_k \in \mathcal{H}}{\operatorname{argmin}} F\left(w_k, \widetilde{D_k}\right) = \frac{1}{|\widetilde{D_k}|} \sum_{i=1}^{|\widetilde{D_k}|} \mathcal{L}\left(g(x_k(i), w_k), y_k(i)\right), \tag{1}$$

where we define

**Model Parameter Space:** $\mathcal{H} \subseteq \mathbb{R}^h$ denotes the parameter space for a learning model, where $h \in \mathbb{N}$ stands as the dimension of the parameter space. The local model of each client $k \in K$ has the parameter $w_k \in \mathcal{H}$.

**Local Dataset:** We assume a horizontally partitioned dataset, where the global dataset $D$ is distributed across $|K|$ clients. Each client $k$ possesses a local dataset $D_k$, containing data points sampled from the global dataset according to a distribution $\psi_k$.

**Noisy Dataset:** For each client $k$, we define the ground truth data distribution as $\tau_k$ and the potentially noisy data distribution as $\rho_k$. Then, a noisy dataset satisfies the following:

$$\Pr_{\tau_k}(y|x) \neq \Pr_{\rho_k}(y|x), \tag{2}$$

where $\Pr$ refers to the probability function for a given distribution and an event. Therefore, we can simply think that the testing dataset is sampled via $\tau_k$, and the training dataset is sampled via $\rho_k$.

**Metamodel:** We define the metamodel as a function $g : \mathbb{R}^{d_x} \times w_k \to \mathbb{R}^{d_y}$. This function describes a trained model that predicts labels for given data features and model parameters. For convenience, we have $g(x_k(i), w_k) = \widehat{y_k(i)}$.

**Loss Function:** The loss function can be described as $\mathcal{L} : \mathbb{R}^{d_y} \times \mathbb{R}^{d_y} \to \mathbb{R} \geq 0$. For example, in our case, we use the cross-entropy loss for each client $k$:

$$\mathcal{L}_k : \left(\widehat{v_k(i)}, v_k(i)\right) \to -\sum_{j=1}^{|C|} v_k(i)(j) \cdot \log\left(\widehat{v_k(i)(j)}\right), \tag{3}$$

where $C$ is the set of classes, $v_k(i)(j)$ is the one-hot probability vector, $v_k(i)$ represents the observed value $y_k(i)$, and $j$ represents the $j$th value in vector $v_k(i)$. $\widehat{v_k(i)(j)}$ is the predicted probability vector, given by: $\widehat{v_k(i)(j)} = \text{Softmax}(f_k(x_k(i)))$, where $f_k$ is the raw output produced by the neural network before being processed by the softmax function.

In contrast to our approach, which explicitly addresses noise by constructing $\widetilde{D_k}$, clients in standard DFL directly minimize the empirical risk function on their raw, potentially noisy dataset $D_k$, which can be represented in the following:

$$\underset{w_k \in \mathcal{H}}{\operatorname{argmin}} F\left(w_k, D_k\right) = \frac{1}{|D_k|} \sum_{i=1}^{|D_k|} \mathcal{L}(g(x_k(i), w_k), y_k(i)), \tag{4}$$

## 4 PROPOSED ALGORITHM : DFLMV

DFLMV is a three-stage DFL training method developed to tackle learning from commonly seen non-malicious label errors, unintentionally generated by workers due to their diverse annotation expertise and carefulness, in datasets owned by distributed entities. DFLMV comprises three stages: initial training stage, label correction stage, and retraining stage, as elaborated below. The pseudocodes of DFL and DFLMV are provided in Appendix B.1 and Appendix B.2, respectively.

### 4.1 INITIAL TRAINING STAGE

DFLMV begins with the traditional DFL, in which each client will use stochastic gradient descent (SGD) on their local dataset $D_k$ for $E$ local epochs to minimize empirical risk function and thus minimize their local training loss. Specifically, each client first needs to get an initial model by doing the gradient descent:

$$\Delta w_k^{T+1} \leftarrow w_k^T - \eta_T \nabla F\left(w_k^T, D_{km(T)}\right), \tag{5}$$

where $\eta$ is the learning rate, and $D_{km(T)}$ stands for the $k$th client's mini-batch in the $T$th epoch.

Once the model has been initialized, each client $k$ broadcasts its parameters $w_k$ to its neighboring clients. Afterward, the client $k$ waits for $n_{peers}$ model parameters to be received, and once it receives the $n_{peers}$ model parameters, it aggregates the models by using the FedAvg algorithm:

$$w_k^T \leftarrow \Sigma_{j=1}^{|K|} \frac{n_j}{n_{peers}} w_j^T. \tag{6}$$

The new aggregated model $w_k^T$ will be trained for $E$ local epochs before it is ready to be broadcast again. During the initial training phase, each client will repeat the above steps until it reaches a stable point (i.e., the loss value of the local model does not decrease).

### 4.1.1 UPPER BOUND ON THE GENERALIZATION ERROR OF DFL

In order to analyze how various noisy training datasets affect the performance of a machine learning model, we consider each data point as a multi-dimensional random variable (RV), denoted by $(X, Y)$, where $X$ represents the feature and $Y$ is the label. Accordingly, a dataset of client $k$ can be represented as a vector of random variables:

$$D_k = \{(X_k(1), Y_k(1)), \dots, (X_k(|D_k|), Y_k(|D_k|))\}, \tag{7}$$

where $(X_k(i), Y_k(i))$ is the $i$th data point in $k$th client's dataset.

We define client $k$'s empirical risk function (given the potential noisy dataset) as:

$$R_k(w_k) = \mathbb{E}_{D_k \sim \rho_k}[\mathcal{L}_k(g(X_k, w_k), Y_k)], \tag{8}$$

where $w_k$ is the model parameter of client $k$; $\mathbb{E}(.)$ is the expectation function. Similarly, we define the client $k$'s ground-truth risk function (given a clean dataset) as:

$$R_k^*(w_k) = \mathbb{E}_{D_k \sim \tau_k}[\mathcal{L}_k(g(X_k, w_k), Y_k)]. \tag{9}$$

Then we follow Yagli et al. (2020) to define client $k$'s generalization error of the given model as:

$$G_k(w_k) = |R_k^*(w_k) - R_k(w_k)|. \tag{10}$$

**Theorem 1 (Upper bound on the generalization error of a given model).** *For any DFL algorithm under label noise that uses the cross-entropy function as the loss function, its generalization error is upper bounded by*

$$G_k(w_k) \le \Omega \cdot \mathbb{E}_{X_k}\left[\sum_{j=1}^{|C|}\left|\Pr_{\rho_k}(Y_k = j|X_k) - \Pr_{\tau_k}(Y_k = j|X_k)\right|\right], \tag{11}$$

*where $\Omega$ is the upper limit among the elements of the vector $f_k$, which is the raw output produced by the neural network before being processed by the softmax function.*

The proof for Theorems 1 is given in Appendix C.1.

**Corollary 1 (Impact of label noise on traditional DFL).** *A lower label noise ratio will result in a lower generalization error $G_k(w_k)$ and a better performance of the trained model.*

The proof for Corollary 1 is provided in Appendix C.2.

### 4.2 LABEL CORRECTION STAGE

During stage two, each client first exchanges model parameters with its neighbors and then uses each neighbor's model to predict a label for each piece of data in its training dataset. Without loss of generality, we denote $B$ as the number of neighbors of a client $k$. We let $Y_j(\widehat{X_k(i)})$ be the predicted label for feature $X_k(i)$ made by client $k$ using its $j$th neighbor's model. Among the $B$ labels made for $X_k(i)$, client $k$ selects the most common one according to majority voting and considers this one as the updated label for $X_k(i)$. Hence, the majority vote protocol can be expressed in the following:

$$Y_k(\widehat{X_k(i)}) = mvf\left(Y_1(\widehat{X_k(i)}), \dots, Y_B(\widehat{X_k(i)})\right) = \underset{z \in C}{\arg\max} \sum_{j=1}^{B} \mathbb{1}(Y_j(\widehat{X_k(i)}) = z), \tag{12}$$

where $Y_k\widetilde{(X_k(i))}$ is the updated label for $X_k(i)$ and $mvf(.)$ is the majority voting function that returns the label that receives the most votes among all $B$ neighbors.

Afterward, client $k$ replaces the original label with the updated label for all the data points in its local training dataset, i.e., client $k$ will do the following:

$$Y_k(i) \leftarrow Y_k\widetilde{(X_k(i))} \quad (\forall X_k(i) \in D_k).$$ (13)

### 4.2.1 Upper Bound on the Error Rate of Majority Voting

Without loss of generality, let us focus our analysis on the majority voting process of the first data point of client 1. Such a treatment allows us to drop the index of the client and the index of data in the analysis and, hence, simplify our presentation. In particular, we denote $A$ as the ground truth classification (i.e., the true label) of the target data point, and we let $\widehat{A_j}$ denote the predicted label for the data point made by the target client by using its $j$th neighbor's model. Then, the discrepancy between $\widehat{A_j}$ and $A$ can be modeled by the conditional probability distribution $\Pr(\widehat{A_j}|A)$. Based on the above simplified notations, we can rewrite the $mvf(.)$ equation as:

$$\widetilde{A} = mvf\left(\widehat{A_1}, \ldots, \widehat{A_B}\right) = \underset{z \in C}{\operatorname{argmax}} \sum_{j=1}^{B} \mathbb{1}(\widehat{A_j} = z),$$ (14)

where $\widetilde{A}$ is the updated label.

We denote $p_{u|r}^{(j)}$ as the probability that the $j$th neighbor's model predicted label $\widehat{A_j} = u$ while $A = r$, where $u, r \in C$, i.e.,

$$p_{u|r}^{(j)} = \Pr\left(\widehat{A_j} = u | A = r\right).$$ (15)

The error rate of the majority voting is defined as:

$$\mathbf{P_e} = \Pr\left(A \neq \widetilde{A}\right).$$ (16)

**Theorem 2 (Upper bound on $\mathbf{P_e}$).** *Given that neighbors are not colluding in their training and that the distributions of the votes are identical, the error rate of the majority voting is upper bounded by:*

$$\mathbf{P_e} \leq 2\left(1 - \frac{1}{|C|}\sum_{r=1}^{|C|}\prod_{\substack{u=1 \\ u \neq r}}^{|C|}\left(1 - \sum_{\beta=0}^{\infty} e^{-B(p_{u|r}+p_{r|r})} \times \left(\frac{p_{u|r}}{p_{r|r}}\right)^{\frac{\beta}{2}} \times I_\beta(2B\sqrt{p_{u|r}p_{r|r}})\right)\right),$$ (17)

*where $\beta$ is an integer, $I_{\beta(.)}$ is the Bessel function of the first kind of order (Mitzenmacher & Upfal, 2017), i.e.,*

$$I_\beta(x) = \sum_{t=0}^{\infty} \frac{(-1)^k}{t!(t+\beta)!}\left(\frac{x}{2}\right)^{2t+\beta}.$$ (18)

We defer the proof for Theorems 2 to Appendix C.3.

**Corollary 2.** *Given that neighbors are not colluding in their training and that the distributions of the votes are identical, the bound on $\mathbf{P_e}$ is monotonically decreasing with $B$. In an extreme case, when $B$ tends to $+\infty$, $\mathbf{P_e}$ tends to 0.*

The proof for Corollary 2 is included in Appendix C.4.

**Corollary 3** *Given that neighbors are not colluding in their training and that the distributions of the votes are identical, higher quality of neighbor's model (i.e., smaller generalization error of the model) reduces $\mathbf{P_e}$.*

We defer the proof for Corollary 3 to Appendix C.5.

### 4.3 Retraining Stage

In this stage, each client will retrain over its updated dataset (denoted as $\widetilde{D_k}$) to fine-tune its model. Specifically, each client will do the gradient descent for $E$ local epochs according to Eq.(5) by using

$\widetilde{D_k}$ and their latest model parameter obtained from Stage 1. Then, the model parameters of each client are passed to all its neighbors. After each client receives all its neighbors' model parameters, it will perform the model aggregation of them according to Eq.(6). The aggregated model $w_j^T$ will then be trained for another $E$ epochs locally before it is exchanged with neighbors again. Upon the completion of the entire retraining stage, each client will get its fully optimized and fine-tuned model $w_k$.

**Theorem 3.** *Define $G_k(w_k^{D_k \sim \rho_k})$ and $G_k(w_k^{\widetilde{D_k} \sim \widetilde{\rho_k}})$ as the generalization error of the models trained by client $k$ over $D_k$ and over $\widetilde{D_k}$, respectively, where $\widetilde{\rho_k}$ is the noisy label distribution in $\widetilde{D_k}$. Given that neighbors are not colluding in their training and that the distributions of the votes are identical, we have*

$$G_k\left(w_k^{D_k \sim \rho_k}\right) > G_k\left(w_k^{\widetilde{D_k} \sim \widetilde{\rho_k}}\right). \tag{19}$$

The proof for Theorems 3 is provided in Appendix C.6.

Theorem 3 states that DFLMV can effectively mitigate the adverse effects of corrupted labels and improve the learned model's accuracy. Based on this observation, it is also easy to see that if the DFL training converges over both the raw dataset $D_k$ and the idealized noise-free dataset (as if there is an Oracle that can assign a true label for every data in the dataset), then DFLMV must converge over the updated dataset $\widetilde{D_k}$. While Theorem 3 assumes non-colluding neighbors with identical vote distributions, the same result, e.g., DFLMV improves model accuracy, is also true under non-IID conditions, as will be verified by our extensive experiment in Section 5.

### 4.4 COMMUNICATION AND COMPUTATION OVERHEAD ANALYSIS

Compared to existing CFL-based label correction methods, which often require additional data processing and the training of auxiliary modules for label correction, leading to increased computation and communication overhead, DFLMV offers significant advantages. DFLMV does not introduce any extra communication overhead, consistent with traditional DFL methods, and the overall communication overhead remains $O(m)$, where $m$ is the number of model parameters exchanged among neighbors. The extra computation overhead introduced by DFLMV is also very low. Specifically, the only additional computation overhead arises in Stage 2, where the majority voting process for updating labels introduces an $O(n)$ computation overhead, where $n$ is the number of data points. A more detailed communication and computation overhead analysis is available in Appendix C.7.

## 5 EXPERIMENTS

In this section, we verify the effectiveness of DFLMV by comparing it with several baseline models across seven datasets under various data/noise settings. Specifically, for each of the synthetic datasets (MNIST (LeCun et al., 1998), Fashion-MNIST (Xiao et al., 2017), and CIFAR-10 (Krizhevsky et al., 2009)), we conduct experiments using a comprehensive set of 12 settings that account for different combinations of IID/non-IID data, IID/non-IID noisy labels, and three different noise models (symmetric, pairflip, and asymmetric). Under each noise model, we consider three different noise ratios: 10%, 30%, and 50%. Additionally, we test DFLMV on four real-world noisy datasets (CIFAR-10N (Wei et al., 2022), CIFAR-100N (Wei et al., 2022), Clothing1M (Xiao et al., 2015b), and ANIMAL-10N (Song et al., 2019)) under non-IID conditions.

### 5.1 EXPERIMENT SETTINGS

**Datasets.** We perform extensive experiments on seven benchmark image datasets, including MNIST, Fashion-MNIST, CIFAR-10, CIFAR-10N, CIFAR-100N, Clothing1M, and ANIMAL-10N. The partitioning of training and testing data for these datasets is summarized in Appendix D.1.

**Generate IID and non-IID Datasets for clients.** To generate disjoint IID datasets for clients, we independently assign each data sample to a client by following a uniform distribution. For the non-IID case, we focus on label distribution skew (Kairouz et al., 2021). In particular, we partition the original training dataset into $K$ disjoint non-IID training datasets via Dirichlet distribution $p_c \sim Dir_K(\alpha)$ (Hsu et al., 2019), where $\alpha \in (0, +\infty)$, and $\alpha$ is the concentration parameter. The smaller

the value of $\alpha$, the greater the level of heterogeneity of the subsets will be. In our case, we chose $\alpha = 1.5$.

**Generate IID and non-IID noisy labels.** Due to the heterogeneous nature of clients, the noisy labels can be not only IID but also non-IID among clients' training datasets (Xu et al., 2022; Görnitz et al., 2014). We consider three different noisy label models: symmetric noise (Figure 1 (a)), pairflip noise (Figure 1 (b)), and asymmetric noise (Figure 1 (c)). Specifically, to generate IID noisy labels in the symmetric noise model, a fraction of data in each class will flip their labels respectively to a randomly selected (wrong) label. In the pairflip noise model, a fraction of data in each class will flip their labels to the label of the next class. In the asymmetric noise model, a fraction of data in two similar classes will swap their labels. In particular, in the CIFAR-10 dataset, a fraction of data in the automobile class swaps label with that in the truck class (denoted as automobile↔truck) and cat↔dog; in MNIST dataset, we have 1↔7 and 0↔6; in Fashion-MNIST dataset, we have T-shirt↔shirt and pullover↔coat. For each noise model, we consider three noise ratios (i.e., the fraction of data that has wrong labels): 10%, 30%, and 50%. To generate non-IID noisy labels, we first generate IID noisy labels for each client based on the aforementioned process. We then collect the data points of wrong labels from all clients and re-assign these data points to clients based on $Dir_K(\alpha = 1.5)$.

Note that we do not introduce extra label noise to CIFAR-10N, CIFAR-100N, Clothing1M, and ANIMAL-10N, as these datasets already contain real-world label noises. Similar to previous studies (Li et al., 2024a), we only consider non-IID data partitions for these real-world noisy datasets.

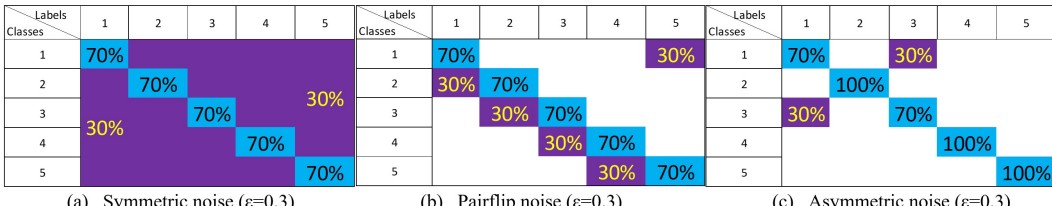

(a) Symmetric noise (ε=0.3)      (b) Pairflip noise (ε=0.3)      (c) Asymmetric noise (ε=0.3)

Figure 1: Noise models ($[C] = 5$ and the noise ratio $\epsilon = 30\%$ in the illustration). A blue (or purple) grid indicates the fraction of data in the class that has correct (or wrong) labels, respectively.

**Baselines and Model Parameters.** We compare DFLMV with FedAVG (Li et al., 2019) and modified PENS (Onoszko et al., 2021). We modified the settings of PENS so that each client sends and receives model parameters from all other online neighbors in order to achieve a fair comparison with DFLMV. Our experiments utilize different hyperparameter settings for various datasets. For MNIST and Fashion-MNIST, we use SGD with 0 momentum, a weight decay of 0.001, a learning rate of 0.01, and a batch size of 50 as the local optimizer. We set 300 global epochs, with 3 local epochs in each global epoch. For CIFAR-10, CIFAR-10N, CIFAR-100N, Clothing1M, and ANIMAL-10N, we use SGD with 0.9 momentum, a weight decay of 0.0005, a learning rate of 0.001, and batch sizes of 32, 16, 16, 10, and 16, respectively, as the local optimizer. We set 400 global epochs for CIFAR-10, CIFAR-10N, and CIFAR-100N, and 200 global epochs for Clothing1M and ANIMAL-10N, with 3 local epochs in each global epoch. The label-correcting parameter is set to 150 for MNIST, 200 for Fashion-MNIST, CIFAR-10, CIFAR-10N, and CIFAR-100N, and 100 for Clothing1M and ANIMAL-10N.

We only use non-pretrained models at the beginning of our experiments. This is because pre-trained models, such as the ResNet families, are trained on large datasets. Using them from the initial step can introduce a potential confound when evaluating the efficacy of our label correction method. Specifically, the improvements observed after label correction could be partially attributed to the pre-existing knowledge embedded in the pre-trained models rather than solely to the correction method itself. Starting with non-pretrained models precludes the impact of the aforementioned bias, allowing a more accurate assessment of the performance gains contributed only by the proposed correction methods. All experiments are executed on Tesla P100 ×16. The details on each network structure are given in Appendix D.2.

### 5.2 EXPERIMENT RESULTS

Experiment results are given in Tables 1 through 5, where for each learning method, we report the average classification accuracy of all clients' models on the 7 benchmark datasets. After observing

| IID datasets and IID noisy label distribution | | | | | | | | | | |
|---|---|---|---|---|---|---|---|---|---|---|
| Dataset | Method | Test Accuracy (%) | | | | | | | | |
| | Noise Type | N/A | Symmetric | | | Pairflip | | | Asymmetric | | |
| | Noise Ratio | 0 | 0.1 | 0.3 | 0.5 | 0.1 | 0.3 | 0.5 | 0.1 | 0.3 | 0.5 |
| MNIST | FedAvg (FL) | 99.21 | 98.32 | 98.09 | 97.67 | 98.7 | 97.81 | 52.7 | 98.13 | 98.01 | 81.01 |
| | PENS (DFL) | 99.35 | 98.4 | 98.14 | 97.23 | 98.2 | 97.78 | 51.35 | 98.21 | 98.11 | 80.93 |
| | DFLMV (DFL) | 99.26 | 98.39 | 98.12 | 97.71 | 98.21 | 97.97 | 60.94 | 98.25 | 98.28 | 84.17 |
| Fashion-MNIST | FedAvg (FL) | 92.5 | 90.01 | 89.11 | 87.41 | 90.07 | 84.1 | 45.55 | 90.12 | 89.21 | 75.7 |
| | PENS (DFL) | 91.56 | 89.91 | 89.09 | 87.38 | 90.03 | 88.6 | 46.17 | 90.1 | 89.35 | 75.3 |
| | DFLMV (DFL) | 91.77 | 89.94 | 89.02 | 87.47 | 90.04 | 89.75 | 51.01 | 90.11 | 89.57 | 75.34 |
| CIFAR-10 | FedAvg (FL) | 71.4 | 58.97 | 56.01 | 46.91 | 61.23 | 45.21 | 32.11 | 63.15 | 60.11 | 56.32 |
| | PENS (DFL) | 71.1 | 59.11 | 55.11 | 47.65 | 61.32 | 46.21 | 32.12 | 63.21 | 60.3 | 56.17 |
| | DFLMV (DFL) | 71.2 | 59.01 | 57.82 | 49.41 | 61.38 | 46.4 | 33.03 | 63.85 | 60.99 | 56.82 |

Table 1: Test accuracy results under IID datasets and IID noisy labels setting.

| IID datasets and non-IID noisy label distribution | | | | | | | | | | |
|---|---|---|---|---|---|---|---|---|---|---|
| Dataset | Method | Test Accuracy (%) | | | | | | | | |
| | Noise Type | N/A | Symmetric | | | Pairflip | | | Asymmetric | | |
| | Noise Ratio | 0 | 0.1 | 0.3 | 0.5 | 0.1 | 0.3 | 0.5 | 0.1 | 0.3 | 0.5 |
| MNIST | FedAvg (FL) | 99.21 | 98.12 | 90.12 | 85.54 | 77.23 | 70.4 | 49.7 | 95.49 | 77.58 | 66.36 |
| | PENS (DFL) | 99.35 | 97.97 | 91.67 | 87.21 | 75.57 | 69.54 | 51.5 | 95.55 | 79.32 | 65.4 |
| | DFLMV (DFL) | 99.26 | 98.31 | 95.47 | 96.35 | 98.3 | 90.78 | 71.71 | 97.6 | 90.04 | 73.78 |
| Fashion-MNIST | FedAvg (FL) | 92.5 | 83.7 | 65.5 | 56.01 | 65.5 | 46.7 | 37.8 | 72.97 | 70.2 | 55.39 |
| | PENS (DFL) | 91.56 | 82.8 | 66.2 | 55.76 | 63.1 | 46.2 | 37.19 | 73.23 | 68.26 | 56.27 |
| | DFLMV (DFL) | 91.77 | 90.09 | 81.7 | 79.04 | 84.85 | 70.12 | 53.49 | 82.48 | 74.81 | 70.65 |
| CIFAR-10 | FedAvg (FL) | 71.4 | 59.61 | 47.25 | 30.21 | 51.27 | 40.11 | 25.23 | 52.42 | 45.55 | 42.61 |
| | PENS (DFL) | 71.1 | 60.12 | 48.31 | 30.11 | 50.93 | 39.67 | 26.24 | 54.68 | 46.26 | 40.56 |
| | DFLMV (DFL) | 71.2 | 61.28 | 52.27 | 35.44 | 59.14 | 50.23 | 31.92 | 60.42 | 55.03 | 52.69 |

Table 2: Test accuracy results under IID datasets and non-IID noisy labels setting.

| Non-IID datasets and IID noisy label distribution | | | | | | | | | | |
|---|---|---|---|---|---|---|---|---|---|---|
| Dataset | Method | Test Accuracy (%) | | | | | | | | |
| | Noise Type | N/A | Symmetric | | | Pairflip | | | Asymmetric | | |
| | Noise Ratio | 0 | 0.1 | 0.3 | 0.5 | 0.1 | 0.3 | 0.5 | 0.1 | 0.3 | 0.5 |
| MNIST | FedAvg (FL) | 99.1 | 96.83 | 95.02 | 94.02 | 93.32 | 89.6 | 55.21 | 92.56 | 90.6 | 75.71 |
| | PENS (DFL) | 98.87 | 96.15 | 95.07 | 94.32 | 93.67 | 88.21 | 56.01 | 93.23 | 90.2 | 76.35 |
| | DFLMV (DFL) | 98.9 | 97.98 | 96.34 | 94.64 | 95.86 | 95.85 | 72.66 | 96.09 | 94.85 | 84.85 |
| Fashion-MNIST | FedAvg (FL) | 92.01 | 70.11 | 67.6 | 63.12 | 73.4 | 69.5 | 42.51 | 72.74 | 71.12 | 65.35 |
| | PENS (DFL) | 91.22 | 70.17 | 68.2 | 64.46 | 72.27 | 70.7 | 43.11 | 73.11 | 71.97 | 65.21 |
| | DFLMV (DFL) | 91.21 | 70.15 | 69.61 | 68.18 | 73.93 | 79.12 | 53.97 | 75.52 | 74.11 | 66.72 |
| CIFAR-10 | FedAvg (FL) | 68.67 | 42.61 | 30.49 | 23.12 | 42.7 | 30.21 | 20.12 | 51.62 | 49.89 | 46.56 |
| | PENS (DFL) | 68.47 | 43.48 | 29.21 | 22.67 | 43.62 | 31.11 | 20.25 | 51.43 | 50.12 | 45.21 |
| | DFLMV (DFL) | 68.66 | 51.11 | 36.41 | 27.38 | 50.83 | 33.59 | 21.29 | 54.69 | 52.79 | 49.81 |

Table 3: Test accuracy results under non-IID datasets and IID noisy labels setting.

| Non-IID datasets and Non-IID noisy label distribution | | | | | | | | | | |
|---|---|---|---|---|---|---|---|---|---|---|
| Dataset | Method | Test Accuracy (%) | | | | | | | | |
| | Noise Type | N/A | Symmetric | | | Pairflip | | | Asymmetric | | |
| | Noise Ratio | 0 | 0.1 | 0.3 | 0.5 | 0.1 | 0.3 | 0.5 | 0.1 | 0.3 | 0.5 |
| MNIST | FedAvg (FL) | 99.1 | 91.5 | 76.4 | 69.97 | 81.3 | 73.2 | 53.25 | 85.37 | 76.3 | 72.1 |
| | PENS (DFL) | 98.87 | 90.6 | 75.6 | 70.01 | 80.46 | 70.49 | 51.11 | 84.32 | 76.32 | 72.22 |
| | DFLMV (DFL) | 98.9 | 97.16 | 94.36 | 96.78 | 95.86 | 85.73 | 76.88 | 96.09 | 90.64 | 83.28 |
| Fashion-MNIST | FedAvg (FL) | 92.01 | 77.57 | 68.45 | 62.67 | 68.2 | 50.1 | 20.6 | 80.81 | 70.5 | 69.21 |
| | PENS (DFL) | 91.22 | 78.61 | 70.27 | 63.61 | 71.3 | 49.6 | 22.2 | 78.51 | 72.5 | 68.71 |
| | DFLMV (DFL) | 91.21 | 83.25 | 84.99 | 82.69 | 81.29 | 61.27 | 34.75 | 93.44 | 77.27 | 83.33 |
| CIFAR-10 | FedAvg (FL) | 68.67 | 47.11 | 41.18 | 31.71 | 35.67 | 31.69 | 18.13 | 50.63 | 48.12 | 42.31 |
| | PENS (DFL) | 68.47 | 45.51 | 40.59 | 30.14 | 36.42 | 31.2 | 17.99 | 49.91 | 47.71 | 41.35 |
| | DFLMV (DFL) | 68.66 | 52.76 | 45.64 | 36.52 | 44.76 | 36.07 | 25.72 | 53.48 | 50.2 | 48.33 |

Table 4: Test accuracy results under non-IID datasets and non-IID noisy labels setting.

these tables, our major insight is that while DFLMV leads to a significant enhancement in the average accuracy of the learning models over its counterparts across the majority of the tested cases, the magnitude of improvement depends on the level of heterogeneity of the datasets (as measured by the distribution of data and the quality of their labeling) owned by different clients.

More specifically, in cases where clients' datasets contain heterogeneous data and noise, DFLMV typically provides significant model accuracy improvement. These are relevant to the situations where the data is non-IID (Table 3), the noisy label is non-IID (Table 2), both the data and the noisy label are non-IID (Table 4), and the real-world scenarios (Table 5). These are also relevant to the situation where even though both data and noisy labels are IID, the noisy label distributions are statistically different across classes (i.e., the "Pairflip" and "Asymmetric" columns in Table 1). In

| Real-World Noisy Datasets (non-IID) | | | |
|---|---|---|---|
| Dataset | Test Accuracy (%) | | |
| | CIFAR-10N (worst) | CIFAR-100N (noisy100) | Clothing1M | ANIMAL-10N |
| Noise Type | Real-world | Real-world | Real-world | Real-world |
| Noise Ratio | 0.402 | 0.402 | 0.38 | 0.08 |
| FedAvg (FL) | 49.12 | 30.28 | 34.93 | 54.1 |
| PENS (DFL) | 50.51 | 29.97 | 35.17 | 55.92 |
| DFLMV (DFL) | 59.56 | 33.11 | 38.33 | 57.79 |

Table 5: Test accuracy results under real-world noisy datasets.

these cases, it can be observed that accuracy improvement brought by DFLMV is apparent, typically ranging from about several percent to over $20\%$: the greater the diversity, the more significant the improvement. For example, under a high noisy label ratio of 0.5, Table 4 shows that an over 23% improvement is achieved by DFLMV over PENS and FedAvg under the Pairflip noise model. This observation on the accuracy gain obtained under non-IID scenarios is not surprising because the majority voting occurs after the convergence of initial training. This design allows each client first to benefit from the vanilla FL learning process, which improves each client's local model accuracy even under non-IID settings (i.e., significant variations in data distribution among clients). More specifically, this initial training ensures that each client's model benefits from shared insights among neighbors while retaining its unique understanding of local data. Consequently, even in the extreme scenario where the training datasets of different clients possess data of different classes, the initial training still helps a client to learn a model that is adapted to the global data distribution (instead of being restricted to the client's local training data). As a result, in the subsequent majority voting, for a given data point, those clients that are making a correct label prediction will point to the same label, while clients making incorrect predictions will likely point to different labels (this is because for the given data point there is only one label to be a correct label but there are many different labels to be wrong label). Therefore, a majority voting mechanism is likely to return the correct label, leading to an improved dataset with fewer labeling errors, and hence higher model accuracy after being trained over this improved dataset during the retraining stage.

On the other hand, in the less-diverse case where both data and noisy labels are IID across datasets and across classes of the same dataset, the improvement achieved by DFLMV is minor, typically ranging from 0 to just a couple percent. This case is relevant to the "symmetric" columns in Table 1. The slight improvement can be explained by noting the fact that even though the distributions of data/noisy-label are IID, their realizations may not be identical – the probability that the same data point is given the same wrong label in two clients' datasets is low. For example, even when the noise ratio is as high as 50%, for the MNIST dataset, the probability that a datapoint is given the same wrong label in two clients' datasets is $(\frac{1}{2})^2 \times (\frac{1}{9})^2 \times 9 = \frac{1}{36}$. As a result, it is unlikely that the same mislabeled data is used for training at two clients. Therefore, in the proposed label correction stage, even if the target client has mislabeled data in its training dataset, there is still a good chance that most of its neighbors' models have been trained on the correct label of the data point, and hence are able to correct the wrong label via majority voting.

## 6 CONCLUSION AND FUTURE WORK

In this paper, we attempt to address the problem of learning from datasets with common non-malicious label errors, which are often present in decentralized ownership due to annotators' lack of expertise or carelessness. We focus on mitigating the adverse effects of corrupted labels when implementing DFL systems. To tackle the issue at hand, we propose a novel method, DFLMV, a general DFL framework that enables all clients to collaborate to address inevitable noisy labels on decentralized data ownership. Particularly, with DFLMV, clients can correct their labels efficiently and cost-effectively while maintaining their local data privacy. In Section 4, our theoretical analysis rigorously demonstrates that DFLMV is capable of correcting noisy labels with high confidence. In Section 5, extensive experiments conducted on 7 benchmark datasets show that our proposed approach is robust against noisy labels and performs well in diverse noise settings and data settings. We note that this paper only considers an unweighted plurality majority vote as the label correction mechanism. There are many other possible voting methods that could be more effective or suitable for different scenarios, such as weighted majority vote and probabilistic vote. In our future work, we plan to explore more sophisticated voting methods to further improve the accuracy of learning from noisy labels. We believe this paper could lead to new directions in handling noisy labels in DFL, especially in improving model robustness against noisy labels in decentralized data ownership.

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

# Appendices

## A  APPLY OUR PROPOSED METHOD TO CFL

To apply DFLMV to CFL, we can adapt its core concepts to the standard FedAVG (Li et al., 2019). Specifically, in CFL, once each client's model stabilizes (i.e., the loss function value reaches stability), the parameter server aggregates the parameters of all local models and broadcasts the latest global and local models to all clients. Each client then uses the latest received models to predict a label for each data point in its local training dataset. Subsequently, each client will update the labels according to the DFLMV majority vote protocol. Except for these two minor changes, the other steps will remain the same as in the standard FedAVG.

# B  PSEUDOCODES

---

**Algorithm 1** Decentralized Federated Learning (DFL)

---

1: **Input:** Learning rate $\eta$, number of global commutation round $E_G$, number of local epochs $E_L$, The set of clients $K$

2: **Each Client Executes:**

3: Initialize: $w_k^0, \forall k \in K$

4: **while** $T < E_G$ **do**

5:      WAIT($\Delta$)

6:      random select peers

7:      Broadcasts its parameters $w_k$ to its neighboring clients

8:      Run OnReceiveModel()

9: **end while**

10: **function** OnReceiveModel($w_i^T$)

11: Save($w_i^T$)

12: **if** number of received models $\geq n_{Peers}$ **then**

13:      Merge saved models by doing $w_k^T \leftarrow \Sigma_{j=1}^{|K|}(\frac{n_j}{n_{peers}} w_j^T)$

14:      Client update the local model $w_j$ by doing $\Delta w_k^{T+1} \leftarrow w_k^T - \eta_T \nabla F\left(w_k^T, D_{km(T)}\right)$, where $D_{km(T)}$ stands for the $k$th client's mini-batch in the $T$th epoch.

15: **end if**

16: **end function**

---

---

**Algorithm 2** Majority Voting based Decentralized Federated Learning (DFLMV)

---

**Input:** Learning rate $\eta$, number of global commutation round $E_G$, number of local epochs $E_L$, the set of clients $K$

2: **In Stage 1, each client Executes:**

Initialize: $w_k^0, \forall k \in K$

4: **while** `loss values keep dropping` **do**

     WAIT($\Delta$)

6:      random select peers

     Broadcasts its parameters $w_k$ to its neighboring clients

8:      Run OnReceiveModel()

**end while**

10: **In Stage 2, client** $j$ ($\forall j \in K$) **Executes:**

**for** $i = 1, 2, ..., |D_j|$ **do**

12:      Correct labels based on Majority Vote protocol by doing $Y_k(i) \leftarrow argmax_{z \in C} \sum_{j=1}^{B} \mathbb{1}(Y_j(\widehat{X_k(i)}) = z)$, where $B$ is the number of neighbors of a client $k$; $Y_j(\widehat{X_k(i)})$ represent the subjective prediction of $j$th client's model for $i$th data point in $D_k$.

**end for**

14: **In Stage 3, each client Executes:**

Use the updated dataset $\widetilde{D_k}$ and their latest model parameter from Stage 1, then follow Algorithm 1 for the remaining epochs to complete the training tasks.

16: **function** OnReceiveModel($w_i^T$)

Save($w_i^T$)

18: **if** number of received models $\geq n_{Peers}$ **then**

     Merge saved models by doing $w_k^T \leftarrow \Sigma_{j=1}^{|K|}(\frac{n_j}{n_{peers}} w_j^T)$

20:      Client update the local model $w_j$ by doing $\Delta w_k^{T+1} \leftarrow w_k^T - \eta_T \nabla F\left(w_k^T, \widetilde{D_{km(T)}}\right)$

**end if**

22: **end function**

---

## C PROOF

### C.1 PROOF OF THEOREM 1

**Proof.** To prove Theorem 1, we first need to derive the expectation of cross-entropy loss. Given $(X_k(i), Y_k(i)) \sim \zeta_k$ and $f_k$. The cross-entropy loss in Eq. (3) can be rewritten as:

$$\mathcal{L}_k\left(\widehat{v_k(i)}, v_k(i)\right) = -\sum_{j=1}^{|C|} v_k(i)(j) \cdot \log\left(\text{Softmax}\left(f_k\left(x_k(i)\right)\right)\right) \tag{20}$$

$$= -\sum_{j=1}^{|C|} v_k(i)(j) \cdot \log\left(\frac{e^{f_k^j(x_k(i))}}{\sum_{q=1}^{|C|} e^{f_k^q(x_k(i))}}\right). \tag{21}$$

, where $f_k$ is the vector of raw outputs from the neural network, the value $e \approx 2.78$; $j$ and $q$ are the $j$th and $q$th entry of the vector $f_k$. Since $v_k(i)(j)$ is the one-hot probability vector, it satisfies the following:

$$v_k(i)(j) = 1 \quad \text{and} \quad v_k(i)(a) = 0 \quad \text{for} \quad a \neq j. \tag{22}$$

Hence, we can further simplify the $\mathcal{L}_k$ by using the Eq.(22):

$$\mathcal{L}_k : \left(g\left(X_k, w_k\right), Y_k\right) = -\log\left(\frac{e^{f_k^j(X_k)}}{\sum_q^{|C|} e^{f_k^q(X_k)}}\right) \tag{23}$$

$$= -\left[f_k^j\left(X_k\right) - \log\left(\sum_q^{|C|} e^{f_k^q(X_k)}\right)\right]. \tag{24}$$

Thus, by using the conditional expectation formula, the expectation of cross-entropy loss can be written as follows:

$$\mathbb{E}\left[\mathcal{L}_k\left(g\left(X_k, w_k\right), Y_k\right)\right] = \sum_{j=1}^{|C|} \Pr_{\zeta_k}\left(Y_k = j\right) \mathbb{E}_{X_k|Y_k=j}\left[\mathcal{L}_k\left(g\left(X_k, w_k\right), Y_k\right)\right] \tag{25}$$

$$= -\sum_{j=1}^{|C|} \Pr_{\zeta_k}\left(Y_k = j\right) \mathbb{E}_{X_k|Y_k=j}\left[f_k^j\left(X_k\right) - \log\left(\sum_q^{|C|} e^{f_k^q(X_k)}\right)\right], \tag{26}$$

By using Eq.(26) and the expansion ideal from Ke et al. (2023), we can get:

$$G_k\left(w_k\right) = |R_k^*\left(w_k\right) - R_k\left(w_k\right)| \tag{27}$$

$$= \left|\sum_{j=1}^{|C|}\left[\int_{X_k} f_k^j\left(x_k\right) d\Pr_{\rho_k}\left(x_k, y_k\right) - \int_{X_k} f_k^j\left(x_k\right) d\Pr_{\tau_k}\left(x_k, y_k\right)\right]\right| \tag{28}$$

$$= \left|\sum_{j=1}^{|C|}\left[\int_{X_k} f_k^j\left(x_k\right)\left(d\Pr_{\rho_k}\left(x_k, y_k\right) - d\Pr_{\tau_k}\left(x_k, y_k\right)\right)\right]\right| \tag{29}$$

$$= \left|\mathbb{E}_{X_k}\left[\sum_{j=1}^{|C|} f_k^j\left(X_k\right)\left(\Pr_{\rho_k}\left(Y_k = j|X_k\right) - \Pr_{\tau_k}\left(Y_k = j|X_k\right)\right)\right]\right| \tag{30}$$

$$\text{RHS of Eq.(30)} \leq \mathbb{E}_{X_k}\left[\sum_{j=1}^{|C|} f_k^j\left(X_k\right)\left|\Pr_{\rho_k}\left(Y_k = j|X_k\right) - \Pr_{\tau_k}\left(Y_k = j|X_k\right)\right|\right] \tag{31}$$

$$\text{RHS of Eq.(31)} \leq \Omega \cdot \mathbb{E}_{X_k}\left[\sum_{j=1}^{|C|} \left|\Pr_{\rho_k}\left(Y_k = j|X_k\right) - \Pr_{\tau_k}\left(Y_k = j|X_k\right)\right|\right]. \tag{32}$$

Hence, we have:

$$G_k(w_k) \le \Omega \cdot \mathbb{E}_{X_k} \left[ \sum_{j=1}^{|C|} \left| \Pr_{\rho_k}(Y_k = j | X_k) - \Pr_{\tau_k}(Y_k = j | X_k) \right| \right]. \tag{10}$$

$\square$

## C.2 Proof of Corollary 1

**Proof.** To prove Corollary 1, we first need to formulate the similarity between $\rho_k$ and $\tau_k$. Let us denote $BC(\rho_k, \tau_k)$ as the Bhattacharyya coefficient (Aherne et al., 1998). Given $\sum_{i=1}^{|C|} \rho_k(i) = 1$ and $\sum_{i=1}^{|C|} \tau_k(i) = 1$, the similarity between $\rho_k$ and $\tau_k$ is measured by the following:

$$\cos(\theta) = BC(\rho_k, \tau_k) = \sum_{i=1}^{|C|} \sqrt{\rho_k(i)\tau_k(i)}, \tag{33}$$

where $\theta$ is the difference between $\rho_k$ and $\tau_k$.

From Eq.(10) and Eq.(33), we can easily infer that a lower noisy ratio leads to closer proximity between $\rho_k$ and $\tau_k$ and a smaller value of $|\Pr_{\rho_k}(Y_k = z | X_k) - \Pr_{\tau_k}(Y_k = z | X_k)|$, thereby resulting in a smaller $G_k(w_k)$. For example, in a special case, if $\rho_k$ and $\tau_k$ are identical, then we have:

$$\cos(\theta) = BC(\rho_k, \tau_k) = \sum_{i=1}^{|C|} \sqrt{\rho_k(i)\tau_k(i)} = \sum_{i=1}^{|C|} \sqrt{\rho_k(i)^2} = 1. \tag{34}$$

Since $\cos(\theta) = 1$, we can get $\theta = 0$. Hence, we have:

$$\left| \Pr_{\rho_k}(Y_k = j | X_k) - \Pr_{\tau_k}(Y_k = j | X_k) \right| = \left| \Pr_{\rho_k}(Y_k = j | X_k) - \Pr_{\rho_k}(Y_k = j | X_k) \right| = 0. \tag{35}$$

By substituting Eq.(35) into Eq.(10), we can get:

$$G_k(w_k) \le \Omega \cdot \mathbb{E}_{X_k} \left[ \sum_{j=1}^{|C|} (0) \right] = 0. \tag{36}$$

Therefore, when the noisy ratio $\to 0$, we can get $G_k(w_k) \to 0$. $\square$

## C.3 Proof for Theorem 2

**Proof.** To prove Theorem 2, we first need to derive the conditional probability distribution of the vote count for a given class. We let $\mathbf{S} = (S_1, S_2, S_3, S_4, \ldots, S_B)$ be a random vector, where $S_i \in [1, B]$ represents the count of votes for a particular class. For instance, the number of votes for class $(u)$ is given by:

$$S_u = \sum_{k=1}^{B} \mathbb{1}(\widehat{A}_k = u). \tag{37}$$

Since the vote distributions are identical, the conditional probability $(p_{u|r}^{(j)})$ is independent of the voter $j$. Thus, for the remainder of the paper, we streamline our notation by discarding the superscript $(j)$ and representing it simply as $(p_{u|r})$. Given $\widehat{A}_j = u$ and $A = r$, by utilizing the multinomial distribution formula, we can express the conditional probability distribution of $\mathbf{S}$ as follows:

$$\Pr(\mathbf{S} = \mathbf{s} | A = r) = \frac{B!}{\prod_{i=1}^{|C|} S_i!} \prod_{u=1}^{|C|} (p_{u|r})^{S_u}, \tag{38}$$

*where* $\sum_{j=1}^{|C|} S_j = B$.

From the above Eq.(38), it is easy to see that the RVs $S_1,\ S_2, \ldots, S_B$ are mutually dependent. This lack of independence makes deriving the error rate extremely difficult. To address this, we adopt the Poisson approximation, inspired by Aeeneh (2023); Mitzenmacher & Upfal (2017). *We define $\widehat{\mathbf{S}} = (\widehat{S_1},\ \widehat{S_2},\ \ldots,\ \widehat{S_B})$ as a vector of RVs; each of the RVs $\widehat{S_i} \in \widehat{\mathbf{S}}$ is independent of each other; $\widehat{S_i}$ follows Poisson distribution, $\widehat{S_i} \sim P(\lambda)$, and $\lambda = B \times (p_{u|r})$. Given that $\widehat{A_j} = u$ and $A = r$, the conditional probability distribution of $\widehat{\mathbf{S}}$ can be rewritten as follows:*

$$\Pr\left(\widehat{\mathbf{S}} = \widehat{\mathbf{s}}\Big|A = r\right) = \prod_{u=1}^{|C|} \Pr\left(\widehat{S_u} = \widehat{s_u}\Big|A = r\right) \tag{39}$$

$$= \prod_{u=1}^{|C|} \frac{e^{-B \times (p_{u|r})}(B \times (p_{u|r}))^{\widehat{s_u}}}{\widehat{s_u}!}. \tag{40}$$

To connect the probability events of $\mathbf{S}$ and $\widehat{\mathbf{S}}$, we define $\varepsilon(\mathbf{S})$ as an event whose probability changes monotonically (either increasing or decreasing) based on the number of participants. Similarly, let $\varepsilon\left(\widehat{\mathbf{S}}\right)$ denote the same event applied to the Poisson case. Leveraging Lemma 1 of Aeeneh (2023) and Corollary 5.11 of Mitzenmacher & Upfal (2017), we can establish the following inequality:

$$\Pr\left(\varepsilon\ (\mathbf{S})\right) \le 2\Pr\left(\varepsilon\ \left(\widehat{\mathbf{S}}\right)\right). \tag{41}$$

To make our upper bound more convincing, we consider the worst-case (The distribution of A is uniform over its domain).

$$\Pr(A = r) = \frac{1}{|C|},\ \forall r \in\ C. \tag{42}$$

Then we can rewrite the error rate $\mathbf{P_e}$ as:

$$\mathbf{P_e} = \Pr\left(\widetilde{A} \ne A\right) \tag{43}$$

$$= 1 - \frac{1}{|C|}\sum_{r=1}^{|C|} \Pr\left(\widetilde{A} = A\Big|A = r\right). \tag{44}$$

Let us continue the proof under the worst scenario that $mvf(.)$ tends to select an incorrect class when it breaks ties.

$$\text{RHS of Eq.(44)} \ \le 1 - \frac{1}{|C|}\sum_{r=1}^{|C|}\Pr\left(\bigcap_{\substack{u=1 \\ u \ne r}}^{|C|} S_u < S_r \Bigg| A = r\right) \tag{45}$$

By Eq.(41), we have

$$\text{RHS of Eq.(45)} \le 2\left(1 - \frac{1}{|C|}\sum_{r=1}^{|C|}\Pr\left(\bigcap_{\substack{u=1 \\ u \ne r}}^{|C|} \widehat{S_u} < \widehat{S_r} \Bigg| A = r\right)\right) \tag{46}$$

$$\text{RHS of Eq.(46)} = 2\left(1 - \frac{1}{|C|}\sum_{r=1}^{|C|}\prod_{\substack{u=1 \\ u \ne r}}^{|C|}\Pr\left(\widehat{S_u} < \widehat{S_r}\Big|A = r\right)\right) \tag{47}$$

Since $\Pr\left(\widehat{S_u} < \widehat{S_r}\Big|A = r\right) = 1 - \sum_{\beta=0}^{\infty}\Pr\left(\widehat{S_u} - \widehat{S_r} = \beta\Big|A = r\right)$, Hence, we have:

$$\text{RHS of Eq.(47)} = 2\left(1 - \frac{1}{|C|}\sum_{r=1}^{|C|}\prod_{\substack{u=1 \\ u \ne r}}^{|C|}\left(1 - \sum_{\beta=0}^{\infty}\Pr\left(\widehat{S_u} - \widehat{S_r} = \beta\Big|A = r\right)\right)\right) \tag{48}$$

By substituting the $\Pr\left(\widehat{S_u} - \widehat{S_r} = \beta \middle| A = r\right)$ with Skellam PMF, the above equation equals:

$$= 2\left(1 - \frac{1}{|C|}\sum_{r=1}^{|C|}\prod_{\substack{u=1\\u\neq r}}^{|C|}\left(1 - \sum_{\beta=0}^{\infty}e^{-B(p_{u|r}+p_{r|r})}\times\left(\frac{p_{u|r}}{p_{r|r}}\right)^{\frac{\beta}{2}}\times I_\beta(2B\sqrt{p_{u|r}p_{r|r}})\right)\right), \quad (30)$$

where, $I_{\beta(.)}$, is the Bessel function of the first kind of order (Dobrushkin, 2017). $\qquad\square$

Based on the above theorem, we propose two corollaries regarding the impact of the number of neighbors (i.e., $B$) and the quality of the neighbor's model (represented by the model's generalization error) on $\mathbf{P_e}$.

C.4  PROOF FOR COROLLARY 2

**Proof.** To prove Corollary 2, we first need to notice the monotonicity of the envelope function of the Bessel function. Specifically, according to Weber et al. (2004), the Bessel function $I_\beta(x)$ exhibits oscillations without periodicity. As $x$ increases, the amplitude of these oscillations decays asymptotically with $x^{-1/2}$, ultimately approaching 0 when $x \to \infty$. If we denote the upper envelope of $I_\beta(x)$ as $env^{upper}I_\beta(x)$, then the trend of $env^{upper}I_\beta(x)$ also remains positive and decreases monotonically as $x$ increases.

To simplify the proof, we use $env^{upper}I_\beta(x)$ to replace $I_\beta(x)$ and analyze the trend of the upper bound of error rate $\mathbf{P_e}$ in relation to $B$. We defined the following:

$$\Xi(B) = \sum_{\beta=0}^{\infty}e^{-B\times(p_{u|r}+p_{r|r})}\times\left(\frac{p_{u|r}}{p_{r|r}}\right)^{\frac{\beta}{2}}\times env^{upper}I_\beta\left(2B\sqrt{p_{u|r}p_{r|r}}\right), \quad (49)$$

Then Eq.(17) can be rewritten as the following:

$$\mathbf{P_e} \leq 2\left(1 - \frac{1}{|C|}\sum_{r=1}^{|C|}\prod_{\substack{u=1\\u\neq r}}^{|C|}(1 - \Xi(B))\right). \quad (50)$$

From Eq.(49), we observe that $e^{-B\times(p_{u|r}+p_{r|r})}$ is positive and decreases monotonically with $B$. Since the product of two positive, monotonically decreasing functions is also a monotonically decreasing function, we can conclude that $e^{-B\times(p_{u|r}+p_{r|r})}\times env^{upper}I_\beta\left(2B\sqrt{p_{u|r}p_{r|r}}\right)$ decreases monotonically with $B$. Consequently, $\Xi(B)$ decreases monotonically with $B$, $1 - \Xi(B)$ increases monotonically with $B$, and the RHS of Eq. (50) decreases monotonically with $B$. Therefore, the bound on $\mathbf{P_e}$ monotonically decreases with $B$.

In an extreme case, when $B \to \infty$, then $2B\sqrt{p_{u|r}p_{r|r}} \to \infty$, $env^{upper}I_\beta\left(2B\sqrt{p_{u|r}p_{r|r}}\right) \to 0$, and $e^{-B\times(p_{u|r}+p_{r|r})} \to 0$. So Eq.(49) can be rewritten as:

$$\mathbf{P_e} \leq 2\left(1 - \frac{1}{|C|}\sum_{r=1}^{|C|}\prod_{\substack{u=1\\u\neq r}}^{|C|}\left(1 - \sum_{\beta=0}^{\infty}0\times\left(\frac{p_{u|r}}{p_{r|r}}\right)^{\frac{\beta}{2}}\times 0\right)\right) \quad (51)$$

$$\text{RHS of Eq.(51)} = 2\left(1 - \frac{1}{|C|}\sum_{r=1}^{|C|}\prod_{\substack{u=1\\u\neq r}}^{|C|}(1-0)\right) = 2\left(1 - \frac{|C|}{|C|}\right) = 0. \quad (52)$$

Hence, when $B \to \infty$, we can get $\mathbf{P_e} \to 0$. $\qquad\square$

C.5  PROOF OF COROLLARY 3

**Proof.** To prove Corollary 3, we first need to express the upper bound of the error rate $\mathbf{P_e}$ in terms of the generalization error $G_k(w_k)$. We observe that models with smaller $G_k(w_k)$, for a given

feature, often exhibit a higher probability of correctly predicting $(\widehat{A_k} = r)$ and a lower probability of incorrectly predicting $(\widehat{A_k} = u, \ (u \neq r))$, which suggests $G_k(w_k) \propto p_{u|r}$. Hence, we denote $p_{u|r} = \varphi \times G_k(w_k)$, where $\varphi \in \mathbb{R}^+$. Then the upper bound of the error rate $\mathbf{P_e}$ of $mvf(.)$ can be rewritten as follows:

$$\mathbf{P_e} \leq 2 \left( 1 - \tfrac{1}{|C|} \sum_{r=1}^{|C|} \prod_{\substack{u=1 \\ u \neq r}}^{|C|} \left( 1 - \sum_{\beta=0}^{\infty} e^{-B\left(\varphi \times G_k(w_k) + p_{r|r}\right)} \left( \tfrac{\varphi \times G_k(w_k)}{p_{r|r}} \right)^{\frac{\beta}{2}} I_\beta \left( 2B\sqrt{\varphi \times G_k(w_k) p_{r|r}} \right) \right) \right), \quad (53)$$

From Eq.(53), we can observe that a smaller $G_k(w_k)$ can contribute to a reduction in the $\mathbf{P_e}$. For the extreme case, if $G_k(w_k) = 0$, then $2B\sqrt{\varphi \times G_k(w_k) p_{r|r}} = 0$; $\left( \tfrac{\varphi \times G_k(w_k)}{p_{r|r}} \right)^{\frac{\beta}{2}} = 0$; $e^{-B \times \left(\varphi \times G_k(w_k) + p_{r|r}\right)}$ is bounded by 1; for the Bessel function part, we have:

$$I_\beta(x) = I_\beta \left( 2B\sqrt{\varphi \times G_k(w_k) p_{r|r}} \right) \tag{54}$$

$$= \sum_{t=0}^{\infty} \frac{1}{t!(t+\beta)} \left( \frac{2B\sqrt{\varphi \times G_k(w_k) p_{r|r}}}{2} \right)^{2t+\beta} \tag{55}$$

$$= \sum_{t=0}^{\infty} \frac{(-1)^k}{t!(t+\beta)!}(0) = 0. \tag{56}$$

Hence, Eq.(53) can be rewritten as:

$$\mathbf{P_e} \leq 2 \left( 1 - \frac{1}{|C|} \sum_{r=1}^{|C|} \prod_{\substack{u=1 \\ u \neq r}}^{|C|} \left( 1 - \sum_{\beta=0}^{\infty} e^{-B \times \left(\varphi \times G_k(w_k) + p_{r|r}\right)} \times 0 \times 0 \right) \right) \tag{57}$$

$$\text{RHS of Eq.(57)} = 2 \left( 1 - \frac{1}{|C|} \sum_{r=1}^{|C|} \prod_{\substack{u=1 \\ u \neq r}}^{|C|} (1 - 0) \right) = 2 \left( 1 - \frac{|C|}{|C|} \right) = 0. \tag{58}$$

Therefore, when $G_k(w_k) \to 0$, we can get $\mathbf{P_e} \to 0$. $\qquad \square$

### C.6 PROOF OF THEOREM 3

**Proof.** By Eq.(33), we denote $\theta_1$ as the difference between $\rho_k$ and $\tau_k$, and $\theta_2$ as the difference between $\widetilde{\rho_k}$ and $\tau_k$. Corollary 2 demonstrates that as the number of voters increases, the error rate $\mathbf{P_e}$ of the $mvf(.)$ decreases, which implies that the decision made by $mvf(.)$ is more accurate than individual choice. Therefore, after using $mvf(.)$ to correct noisy labels, we have $\theta_1 > \theta_2$.

After plugging this analysis into RHS of Eq.(10), we can get the following:

$$\mathbb{E}_{X_k} \sum_{z=1}^{|C|} \left| \Pr_{\rho_k}(Y_k = z|X_k) - \Pr_{\tau_k}(Y_k = z|X_k) \right| > \mathbb{E}_{X_k} \sum_{z=1}^{|C|} \left| \Pr_{\widetilde{\rho_k} k}(Y_k = z|X_k) - \Pr_{\tau_k}(Y_k = z|X_k) \right| \tag{59}$$

Hence,

$$\Omega \cdot \mathbb{E}_{X_k} \sum_{z=1}^{|C|} \left| \Pr_{\rho_k}(Y_k = z|X_k) - \Pr_{\tau_k}(Y_k = z|X_k) \right| > \Omega \cdot \mathbb{E}_{X_k} \sum_{z=1}^{|C|} \left| \Pr_{\widetilde{\rho_k} k}(Y_k = z|X_k) - \Pr_{\tau_k}(Y_k = z|X_k) \right| \tag{60}$$

Therefore, we can get:

$$G_k \left( w_k^{D_k \sim \rho_k} \right) > G_k \left( w_k^{\widetilde{D_k} \sim \widetilde{\rho_k}} \right). \tag{61}$$

$$\square$$

### C.7 ANALYSIS OF COMMUNICATION AND COMPUTATION OVERHEAD

**Communication Overhead Analysis.** Our method doesn't introduce any communication overhead, consistent with traditional DFL methods. In Stage 1, clients train their local models independently, with no additional communication required beyond standard model parameter exchanges. In Stage 2, clients exchange model parameters with their online neighbors, a typical operation in DFL that does not introduce extra communication overhead. In Stage 3, clients fine-tune their local models based on the updated dataset without requiring additional communication. Overall, the communication overhead remains $O(m)$, where $m$ is the number of model parameters exchanged among neighbors.

**Computation Overhead Analysis.** The computation overhead of our method is also minimal. In Stage 1, the computational cost is equivalent to traditional DFL, as clients train their local models on their original datasets. In Stage 2, the majority voting process to update labels introduces an $O(n)$ computation overhead, where $n$ is the number of data points. This is a straightforward operation and does not significantly increase the computational burden. In Stage 3, the extra training epochs for fine-tuning the local models are based on the updated dataset, which is necessary for improving model accuracy. This stage has the same computational cost as the initial training stage and does not introduce additional overhead compared to other existing label correction methods. In summary, the introduced computational overhead is (O(n)).

**Comparison with Other Methods.** Compared to other label correction methods, our approach has the following advantages: (1) Other methods often require additional data processing and training of auxiliary modules, increasing computational overhead. Our method streamlines this by directly utilizing the results from Stage 1 in subsequent stages. (2) Other methods may require exchanging additional information with a central server during the label correction pre-processing stage, increasing communication overhead. Our method avoids this by not introducing any additional communication overhead, consistent with traditional DFL methods. For instance, in 'FedCorr' by Xu et al., 2022, each iteration of the label correction pre-processing Stage involves all clients calculating the local intrinsic dimensionality (LID) score and per-sample loss for their current local model, which adds computational load. Additionally, the LID score will also be transmitted to the server during each iteration, further contributing to communication overhead. Similarly, 'CLC' by Zeng et al., 2022, mandates clients to calculate a threshold $c_t$ in each training iteration of their label correction pre-processing stage to determine the global threshold $c_t^G$, again intensifying the computational overhead.

## D EXPERIMENT SUPPLEMENTARY MATERIALS

### D.1 SUMMARY OF OUR USED DATASETS

|               | #Training | #Testing | #Classes | Size of each sample |
|---------------|-----------|----------|----------|---------------------|
| MNIST         | 60,000    | 10,000   | 10       | 28x28x1             |
| Fashion-MNIST | 60,000    | 10,000   | 10       | 28x28x1             |
| CIFAR-10      | 50,000    | 10,000   | 10       | 32x32x3             |
| CIFAR-10N     | 50,000    | 10,000   | 10       | 32x32x3             |
| CIFAR-100N    | 50,000    | 10,000   | 100      | 32x32x3             |
| Clothing1M    | 1,000,000 | 10,000   | 14       | 224x224x3           |
| ANIMAL-10N    | 50,000    | 5,000    | 10       | 64x64x3             |

Table 6: Summary of datasets and their partitioning in the experiments.

### D.2 NETWORK STRUCTURES

| Layer (type) | Output Shape    | Param # |
|--------------|-----------------|---------|
| Conv2d-1     | [-1, 10, 24, 24]| 260     |
| Conv2d-2     | [-1, 20, 8, 8]  | 5020    |
| Dropout2d-3  | [-1, 20, 8, 8]  | 0       |
| Linear-4     | [-1, 50]        | 16050   |
| Linear-5     | [-1, 10]        | 510     |

Table 7: MNIST Network Structure.

| Layer (type) | Output Shape | Param # |
|---|---|---|
| Conv2d-1 | [-1, 32, 28, 28] | 320 |
| Conv2d-2 | [-1, 64, 12, 12] | 18496 |
| Dropout2d-3 | [-1, 64, 12, 12] | 0 |
| Linear-4 | [-1, 600] | 1383000 |
| Linear-5 | [-1, 120] | 72120 |
| Linear-6 | [-1, 10] | 1210 |

Table 8: Fashion-MNIST Network Structure.

| Layer (type) | Output Shape | Param # |
|---|---|---|
| Conv2d-1 | [-1, 64, 16, 16] | 1,792 |
| MaxPool2d-2 | [-1, 64, 8, 8] | 0 |
| Conv2d-3 | [-1, 192, 8, 8] | 110,784 |
| MaxPool2d-4 | [-1, 192, 4, 4] | 0 |
| Conv2d-5 | [-1, 384, 4, 4] | 663,936 |
| Conv2d-6 | [-1, 256, 4, 4] | 884,992 |
| Conv2d-7 | [-1, 256, 4, 4] | 590,080 |
| MaxPool2d-8 | [-1, 256, 2, 2] | 0 |
| Linear-9 | [-1, 4096] | 4,198,400 |
| Linear-10 | [-1, 4096] | 16,781,312 |
| Linear-11 | [-1, 10] | 40,970 |

Table 9: CIFAR-10 Network Structure.

| Dataset | Structure |
|---|---|
| CIFAR-10N | ResNet18 (Non-pretrained ) |
| CIFAR-100N | ResNet18 (Non-pretrained ) |
| Clothing1M | ResNet50 (Non-pretrained ) |
| ANIMAL-10N | ResNet18 (Non-pretrained ) |

Table 10: Network Structure for Real-World Noisy Datasets .

