# OpenReview forum: "Decentralized Federated Learning Over Noisy Labels: A Majority Voting Method"
_ICLR.cc/2025/Conference — ICLR 2025 Conference Withdrawn Submission_

### Official Review · Reviewer_nwPN · 2024-10-25

**Soundness:** 1
**Presentation:** 1
**Contribution:** 2
**Rating:** 3
**Confidence:** 4

**Summary:**

This paper tackles the challenge of training machine learning models in decentralized environments with noisy label data. To address this, the paper proposes a majority voting-based algorithm called DFLMV (Decentralized Federated Learning Majority Voting), designed to enhance model quality in the presence of label noise without the need for a central server. DFLMV consists of three key stages:

**Initial Training Stage:** Each client performs traditional decentralized learning on its local dataset (an merging periodically with their neighbors) to train an initial model. After training for several epochs, each client reaches a « stable loss » value, at which point it moves to the label correction phase.

**Label Correction Stage:** Clients share model parameters with their neighbors, who then use these models to predict labels on their data. For each data point, the client assigns a new label based on majority voting among the predictions from its neighbors, thus correcting potentially noisy labels.

**Retraining Stage:** With updated labels, clients further fine-tune their local models on this improved dataset to enhance model accuracy and robustness.

The paper provides some theoretical analysis simplified results on the performance of DFLMV. It also compares DFLMV to PENS, another noisy label decentralized learning approach, by evaluating both methods across multiple datasets (e.g., MNIST, Fashion-MNIST, and CIFAR-10) under various levels of label noise and data heterogeneity.

**Strengths:**

The paper addresses a meaningful topic, and further exploration in this direction holds strong potential to benefit the research community. The core idea is clear and straightforward, which is an advantage. Furthermore, provided the results withstand scrutiny, it offers a practical and effective approach to improving the robustness of machine learning algorithms against label noise.

**Weaknesses:**

I found the paper challenging to follow, as aspects such as the organization, technical writing, clarity of results, and integration of related work could significantly benefit from improvements. Strengthening these areas would greatly enhance the paper’s overall clarity and impact. To be more precise I provide a specific list of concerns below.

## Quality of Technical Writing
In my view, the level of technical writing in this paper currently falls short of the standards expected for a top-tier conference like ICLR. Here are some specific points that I believe limit its clarity:

**Problem Presentation**: The problem setup presents an optimization problem in Equation (1) before defining the relevant notations. Generally, it would improve readability to first introduce the notations, clearly defining spaces and terms, and then present the optimization problem.

**Unexplained Concepts and Notation**: Several concepts and notations are introduced without sufficient explanation. For example, symbols like $d_x$ and $d_y$ first appear in line 182 without clarification, similarly $f_k$ in line 193, and the term "non-colluding neighbors" in line 299. Although these may appear minor, they contribute to a challenging reading experience, especially in the preliminaries.

**Inconsistencies in Notations**: There are several inconsistencies in the notations, which make the paper harder to follow. Here is a non-exhaustive list:

   - The loss function is first introduced as $\mathcal{L}$ in Equation (1), then $\mathcal{L}_k$ in Equation (3), and switches back to $\mathcal{L}$ in Equation (4) before reverting to $\mathcal{L}_k$ in Equation (8).

   - The output space and corresponding vectors vary between $\mathbb{R}^{d_y}$ and {$1, \dots, C$}, leading to errors such as computing the one-hot encoding of a vector from $\mathbb{R}^{d_y}$ in line 191. This inconsistency also makes Theorem 1 difficult to interpret, as the left-hand side assumes vectors in $\mathbb{R}^{d_y}$ while the right-hand side assumes $Y$ is discrete.

   - Datasets are defined as sets of lowercase samples in Section 3, then as sets of uppercase random variables in Section 4.

   - Both $B$ and $n_{\text{peers}}$ appear to serve the same role, but they are used interchangeably throughout the paper.

## Presentation of the algorithms

The algorithm’s presentation lacks clarity and structure in the main paper. The analysis is intertwined with the algorithm's steps, making it difficult to follow. I would suggest dedicating one section solely to the algorithm's presentation and a separate section for the analysis to improve readability.

## Clarity and soundness of the results

Due to the clarity issues mentioned earlier, the presented results exhibit similar concerns on clarity. While I have reservations about their overall soundness, I believe that at least Theorem 1, as it currently stands, is factually incorrect. It appears that the paper attempts to adapt results from (Ke et al., 2023), but it overlooks key assumptions from the original work, particularly Assumption 5 and Assumption 6. As a result, the conclusions drawn in this paper cannot be valid, and consequently, the proof that follows the steps of (Ke et al., 2023) is also flawed.

## Related work and comparison with existing solutions

The analysis of related work could be strengthened significantly. For instance, in line 125, the claim that decentralized (federated) learning was introduced by Lalitha et al. (2018) is misleading. The research on decentralized algorithms actually has a rich history (see e.g.,https://arxiv.org/pdf/2006.13838).

Additionally, I found the reference to (Yagli et al., 2020) regarding the definition of generalization somewhat confusing. After reviewing that reference, it does not appear to directly address the concept of noisy labels, and I was unable to locate a definition comparable to Equation (10). Could you provide a more appropriate reference for your definition of generalization error, or explain how you derive your definition from Yagli et al. (2020) if it is indeed relevant?

I also believe that the paper would benefit from a comparison with solutions that address stronger forms of corruption, such as Byzantine attacks in decentralized learning (see e.g., https://proceedings.mlr.press/v202/farhadkhani23a/farhadkhani23a.pdf). While I understand that your paper does not focus on these stronger attacks, exploring whether existing defenses in the literature could also apply to the seemingly simpler problem of noisy labels would be valuable. It’s possible that these methods may be less effective in terms of accuracy, but gaining insight on this would be beneficial before developing alternative approaches.

Finally, I would like to note that it appears the current paper is heavily inspired by (Ke et al., 2023), including similarities in notation and results. Given this, it seems somewhat unexpected that (Ke et al., 2023) is cited only once and solely in the appendix. A more thorough engagement with this source, discussing more explicitly how the submission builds upon or differs from Ke et al. (2023) (highlighting both similarities and key differences) could enhance the paper's depth and contextualization.

**Questions:**

Please comment on the above concerns

---

> ### Author Response · Authors · 2024-11-15
> **Response to Reviewer nwPN**
>
> **Response to  Comments: Unexplained Concepts and Notation, Inconsistencies in Notations**
>
> Thank you for your comments. First, we wish to point out that in mathematical contexts, terms like $d_x$ and $d_y$ in $R^{d_x}$ and $R^{d_y}$ are standard notations representing the dimensions of the feature space and label space, respectively. These notations are commonly used in machine learning literature and are familiar to readers within this field. Regarding the term "non-colluding neighbors," it generally implies that the neighbors operate independently without sharing information or coordinating their actions, as is customary in decentralized learning contexts. Additionally, $f_k$ has already been explicitly defined in our manuscript (line 252) as the raw output produced by the neural network before being processed by the softmax function.
>
> **Response to Inconsistencies in Notations**
>
> 1. Thank you for your comments. In our paper, $L$ stands for transient loss function, and $L_k$ is the final loss function. Specifically,
>  $L$ is associated with the on-going training model and used specifically in Equations (1) and (4) during the DFL training process, where the loss is dynamically adjusted by the updated model parameters in each communication round. one the other hand $L_k$ is associated with the locally finally trained model for each client $k$, as exemplified in Equation (8)
>
> 2. Thank you for pointing out our output notation dimension issues. The accurate notation for output space dimension would be $[0,1]^{dy},$ but this error is easy to correct and does not impact the validity of our theorem proofs, as we have consistently treated the output space dimension as $[0,1]^{dy}$ throughout our analyses.
>
> 3. Thank you for your comments. In our manuscript, we intentionally distinguished between realizations, denoted by lowercase letters, and random variables, represented by uppercase letters. This usage is consistent with standard statistical and machine learning literature, where using random variable typically signifies a generalized or abstract representation of data sets, often conceptualizing data points as outcomes from a defined probability distribution.
>
> 4. We introduce $ B $ to denote the total number of models that an individual client could use during the majority voting process. In contrast, $ n_{\text{peer}} $ refers to the actual number of online neighbors that a client can interact with at each communication round, which reflects real-time network conditions and client availability
>
> **Response to the concern about overlooking key assumptions from the original work, and correctness.**
>
> Thank you for your comments. First, It is essential to clarify that in Ke et al., Assumptions 5 (bounded input space) and 6 (bounded model output) are not used in the proof of their Theorem 3, which establishes the  generalization error bounds under centralized FL settings, but rather specify values for the bounding constant $\Omega$ in their Corollary 8, which is irrelevant to the the proof of their Theorem 3.
>
> Furthermore, we wish to point out that both our work and Ke et al., 2023 draw significant inspiration from Yagli et al., 2020, which delineates expected generalization errors for models trained on datasets from different distributions (on their page two). This foundational work can naturally extend to models trained on noisy datasets, as noisy and clean datasets inherently represent different sampling distributions.  Notably, Ke et al. explicitly reference this methodology in their Equation 6, paralleling our use in Equation 10 to define generalization error in decentralized federated learning settings.
>
> **Response to the concern about who come up DFL concept first.**
>
> Thank you for your comments. Our claim, that decentralized learning was introduced by Lalitha et al., 2018, was specifically intended to acknowledge their notable contributions to the field of DFL rather than to claim the origination of all decentralized algorithms. This acknowledgment is widely accepted within the DFL community, as evidenced by multiple authoritative sources. For instance, the comprehensive survey 'Decentralized Federated Learning: A Survey and Perspective (2024)' specifically credits Lalitha et al., 2018, as pioneers in proposing the DFL concept."

---

### Official Review · Reviewer_dUyj · 2024-11-02

**Soundness:** 3
**Presentation:** 3
**Contribution:** 2
**Rating:** 5
**Confidence:** 3

**Summary:**

Noisy labels exist in widely used datasets, which can adversely affect model training. Existing methods aiming for learning with noisy labels mostly focus on centralized learning (CL) and centralized parameter server-based FL (CFL) settings, which cannot be efficiently employed in the decentralized federated learning (DFL) setting.
This paper then proposes a framework for learning with noisy labels in the DFL setting. The proposed framework is made up of three stages: typical DFL training, label correction which is based on majority voting, and extra fine-tuning.
The authors also provide a theoretical upper bound on the generalization error of any DFL algorithm using cross-entropy loss under arbitrary label noise, and an upper bound on the error rate of majority voting.
Experimental results further show the effectiveness of the proposed framework against noisy labels.

**Strengths:**

- The motivation is clear, and the paper is well-written.
- The proposed framework is simple yet effective, and it includes a theoretical analysis of performance bounds.
- Extensive experiments across various IID and non-IID data/noise settings demonstrate the effectiveness of the proposed framework.

**Weaknesses:**

- The assumption that the distributions of the votes are identical seems too strong, particularly under non-IID settings which mostly cannot stand.
- It is still not clear about the extra overhead. Regarding the claim "DFLMV does not introduce any extra communication overhead", stage 2 involves communication among clients, which would introduce extra communication overhead. Besides, stages 2 and 3 can be considered as the extra steps as stage 1 can be viewed as a whole typical DFL training (until it reaches a stable point), aside from the dimension of model parameters, then the extra overhead would be related to the number of iterations and local epochs in the later 2 stages.

**Questions:**

- Regarding stage 2, can the authors please explain why not consider the aggregated model for label correction? Also, in this way, existing methods could be adopted in DFL as well, and the computational cost may vary.
- Can the authors please provide specific extra computational costs, at least for one setting?

---

> ### Author Response · Authors · 2024-11-15
> **Response to Reviewer dUyj**
>
> **Response to Comment:  The assumption that the distributions of the votes are identical seems too strong**
>
> Thank you for your comments regarding the assumption of identical vote distributions in our paper. It seems that there might have been a misunderstanding about the purpose of the assumption made in the paper. The assumption of identical vote distributions is made in the paper just to support our derivation of the closed-form performance bounds of the DFLMV scheme in the special case of IID data and IID label noise. These performance bounds are tractable in the IID case but become intractable in the more general non-IID cases. Therefore, in the paper, we are just doing our best to present whatever theoretical performance bounds that are possible to get and the particular conditions under which these bounds hold so that readers can get a peek into the performance guarantees that can be provided by the proposed scheme under these special (IID) conditions. Although these bounds are under identical vote distributions assumptions, it is still reasonable to speculate that these bounds can serve as conservative estimates when more complex non-IID conditions are concerned. This speculation is supported by the fact that increased data diversity will enable each client to better exploit the contributions of peers in a non-IID environment, thus improving the performance of our approach. This speculation is also validated in Section 5, where our scheme exhibits significant accuracy gains in non-IID conditions compared to IID scenarios. We believe this is also a commonly accepted approach in machine learning (ML) literature to studying the performance of a new ML model.
>
> **Response to Questions: 1**
>
> Thank you for your question concerning the use of an aggregated model for label correction. First, it is important to clarify that in decentralized federated learning, particularly with non-IID data distributions, directly utilizing an aggregated model ( via methods like FedAvg) can significantly impair performance.
>
> This is because averaging parameters from models trained on diverse datasets typically fails to capture each model's differing statistical characteristics. Averaging method usually works well under homogeneity assumption, but it simply does not exist in non-IID settings, resulting in aggregated model parameters that do not accurately represent any specific dataset's unique features. Consequently, such parameters are often ineffective, leading to a model that performs poorly across all participating clients' data. This speculation is also supported by many existing works. For instance, the study "Enhancing One-Shot Federated Learning Through Data and Ensemble Co-Boosting (2024)" illustrates that even on clean datasets, applying FedAvg just once under high non-IID conditions (with $\alpha = 0.05$) on the CIFAR-100 dataset can lead to a dramatic decrease in accuracy to just 6.45\%.
>
> Moreover, adapting existing methods to DFL may not be appropriate. Specifically, in CFL, robust learning algorithms typically rely on a stable server-client connection to manage and dynamically adjust to each client’s model contributions by leveraging auxiliary information like parameter changes. For instance, the approach proposed in "FedCorr: Multi-Stage Federated Learning for Label Noise Correction (2022)" requires clients to compute and share Local Intrinsic Dimensionality (LID) scores with a central server, which uses these scores to categorize clients into clean or noisy groups via a Gaussian Mixture Model (GMM). These method are infeasible in DFL settings due to the loose network connections and limited client-to-client communications, which restrict access to comprehensive, real-time information across all clients. These restrictions directly hinder clients from effectively acquiring the real-time auxiliary information needed for such label correction methods. Furthermore, CL robust learning methods usually require access to the entire dataset for effective label correction. However, in DFL, each client’s data remains localized, precluding the comprehensive dataset access necessary for such robust learning method.
>
> **Response to Questions: 2**
>
> We explicitly address the additional computational costs in Section 4.4 of our paper, which primarily refer to operations beyond standard training routines, such as gradient descent, loss calculation, and parameter updates. For instance, the specific additional costs in our decentralized federated learning (DFL) framework occur in Stage 2 during the majority voting process for label updates. In this stage, each client uses all received models to make predictions on each data point within its local dataset. Following this, a majority voting mechanism is employed to update the labels of each data point. This process introduces an $O(n)$ computational overhead, where $n$ is the number of data points in a client’s dataset.

---

### Official Review · Reviewer_Fgzx · 2024-11-04

**Soundness:** 2
**Presentation:** 3
**Contribution:** 2
**Rating:** 5
**Confidence:** 3

**Summary:**

This paper introduces DFLMV, a decentralized federated learning (DFL) algorithm for training under noisy labels. DFLMV has in total three stages: (1) initial model training with traditional DFL, (2) label correction using a majority voting mechanism with peer models, and (3) fine-tuning models on the corrected data. This approach tackles the label noise issue without a centralized server and without the need for additional clean datasets. The authors provided theoretical guarantees on generalization error and error rate bounds for the majority voting mechanism. In addition, the authors conducted extensive experiments to demonstrate the accuracy improvements of DFLMV across several benchmark datasets.

**Strengths:**

- The idea of using peer model majority voting in the DFL setting for handling label noise is novel.
- The paper provided clear explanations of the stages in DFLMV with the detailed theoretical analysis for each component of the algorithm.
- The evaluation of DFLMV is extensive where the authors considered different data distribution and label noise distribution. The results all demonstrated that the proposed method is effective.

**Weaknesses:**

Some details and ablation studies about the dataset partition are missing: how many clients are there for each dataset? For each client, how are the neighbors defined? How does the number of neighbors affect the utility of this approach? How many data points are needed for the initial staging to provide a reasonable model for relabeling?

**Questions:**

- How robust is this approach considering malicious clients who might intentionally send manipulated model parameters to mess up the majority voting process?
- Have you considered other voting methods, e.g. weighted majority vote based on peer distance?

---

> ### Author Response · Authors · 2024-11-15
> **Response to Reviewer Fgzx**
>
> **Response to Comment 1:**
>
> Thank you for your question regarding the robustness of our approach against malicious clients within decentralized federated learning (DFL). First, it is crucial to clarify that this work primarily addresses common, real-world scenarios where label errors are non-malicious, often emerging unintentionally due to annotators' lack of knowledge or oversight. As such, our focus is not on adversarial settings where clients might collude to deliberately manipulate model parameters.
>
> Nevertheless, it is important to note that our implementation of the majority voting mechanism does inherently offer a layer of protection against such manipulative behaviors. By aggregating model updates from a majority of participants, this method naturally dilutes the impact of any erroneous or intentionally skewed updates from a minority of clients. This aggregation acts as a fundamental safeguard, ensuring that as long as the majority of clients are honest, the influence of malicious activities can be significantly mitigated.
>
> **Response to Comment 2:**
>
> Thank you for your question regarding the exploration of alternative voting methods in our decentralized federated learning framework. In this study, we intentionally focused on assessing the effectiveness of a straightforward majority voting mechanism to establish a baseline for its efficacy in decentralized federated learning environments. As shown in Section 5, our experimental results demonstrate that this method significantly improves accuracy and generality
>
> in both IID and non-IID data/label-noise settings, indicating that even a simple majority voting method could significantly enhance model robustness and reliability. Based on these encouraging findings, we are planning to explore more sophisticated voting strategies, such as weighted majority voting based on peer distance, in future work.

---

### Official Review · Reviewer_VyNM · 2024-11-04

**Soundness:** 1
**Presentation:** 2
**Contribution:** 2
**Rating:** 3
**Confidence:** 4

**Summary:**

The paper presents a decentralized federated learning algorithms to handle noisy training labels. The federated networks first perform decentralized training. Then each client makes predictions based on the majority voting from the models from its neighbours. The submissions tries to give some theoretical justifications of the proposed approach. Experiments show that the proposed algorithm improves model accuracy across various datasets, under non-IID settings and various noise conditions.

**Strengths:**

Handling noisy training data and understanding the generalization of robust learning algorithms (with distribution shifts) is an important problem.

The proposed method is relatively lightweight, communication efficient, and easy to implement.

Experiments consider various noisy ratios and settings.

**Weaknesses:**

It is not clear what implications Theorem 1 (the generalization result) have. In addition, it is modeling the gap between the population error under two distributions, not involving empirical error over the noisy dataset, making it not applicable to practical settings where we only have access to a finite set of samples. Consequently, Theorem 2 has similar issues.

Clients train on non-IID data, and it is not clear why assuming vote distributions are identical reasonable.

The proof of Theorem 3 is confusing. theta_1 (and theta_2) are the differences between the two probability distributions, but `difference’ is not defined. How does the submission mean by theta_1 is larger than theta_2?

Experiments use simple and similar datasets (despite including a set of image benchmarks). The non-IID dataset partition among clients is not natural partitions, and doesn’t fully reflect the real-world non-IIDness.


Experiments don’t compare with algorithms that target at label noise, for instance, simple baselines such as applying state-of-the-art robust learning algorithms (against noisy labels) locally on each client during decentralized training.

**Questions:**

Please see 'weaknesses'.

---

> ### Author Response · Authors · 2024-11-15
> **Response to Reviewer dUyj**
>
> **Response to Comment 1:**
>
> We wish to point out that there may have been a misunderstanding about the scope and intent of Theorem 1 and 2. The purpose of Theorem 1 is to model the expected performance of a ML model trained on noisy dataset under decentralized DFL settings. In our paper, Theorem 1 does indeed address the empirical risk on noisy datasets, not just the theoretical gap between population errors under two random distributions. Specifically, in our Theorem 1, the generalization error upper bound is derived based on Equation (10), which synthesizes results from Equations (8) and (9). These two functions define the empirical risk functions for a model trained on noisy and clean datasets, respectively.
>
> Furthermore, Theorem 2 builds upon the foundation set by Theorem 1, assessing the effectiveness of a majority voting mechanism in mitigating prediction errors in a decentralized setting, ensuring the relevance of our theoretical findings in practical applications.
>
> **Response to Comment 2:**
>
> The assumption of identical vote distributions made in the paper is just to  support our derivation of closed-form performance bounds for the DFLMV scheme in the special case of IID data and IID label noise. These performance bounds are tractable in the IID case but become intractable in the more general non-IID cases. Therefore, in the paper, we are just doing our best to present whatever theoretical performance bounds that are possible to get and the particular conditions under which these bounds hold so that readers can get a peek into the performance guarantees that can be provided by the proposed scheme under these special (IID) conditions. Although these bounds are under IID assumptions, it is still reasonable to speculate that these bounds can serve as conservative estimates when more complex non-IID conditions are concerned. This speculation is supported by the fact that increased data diversity will enable each client to better exploit the contributions of peers in a non-IID environment, thus improving the performance of our approach.
>
> **Response to Comment 3:**
>
> We wish to point out that you may have misunderstood the term "differences."  The term 'difference' in our context does not refer to a quantitative comparison where one value is numerically greater than the other. Instead, it refers to the degree of divergence between two probability distributions and are calculated by using Bhattacharyya coefficient. In our paper, these measures are explicitly defined in Equation (33) (in the proof of Corollary 1), where we thoroughly outline how these differences between probability distributions are calculated and demonstrate that a lower noise ratio leads to a reduced divergence, indicating a closer proximity between the distributions. In the case mentioned in your comment would be after the label correction stage, the difference between the updated $\widetilde{\theta_1}$ and $\widetilde{\theta_2}$ is smaller than  the original $\theta_1$ and $\theta_2$.
>
> **Response to Comment 4:**
>
> Our experimental evaluation encompasses seven diverse datasets: MNIST, Fashion-MNIST, CIFAR-10, CIFAR-10N, CIFAR-100N, Clothing1M, and ANIMAL-10N. These datasets vary in size, scale, and subject matter. Additionally, our non-IID data partitioning is based on the Dirichlet distribution, a method commonly used to simulate real-world data distribution scenarios.
>
> **Response to Comment 5:**
>
> Thank you for your feedback on our experimental baselines. Indeed, we did not compare our method with existing robust learning algorithms designed for noisy labels. This is because no existing robust learning method can be directly applied in decentralized federated learning (DFL) settings.
>
> Specifically, robust learning algorithms in centralized federated learning (CFL) typically depend on a stable server-client connection, allowing the server to receive real-time auxiliary information from each client. This enables the server to manage and dynamically adjust each client's model contributions effectively. However, in DFL settings, such stable server-client connection doesn't exist due to the fact that the devices in DFL are usually under loose network connections and limited client-to-client communications. Furthermore, CL robust learning methods usually require access to the entire dataset for effective label correction. However, in DFL, each client’s data remains localized, precluding the comprehensive dataset access necessary for such robust learning method.

---

### Note · Authors · 2024-11-24

I have read and agree with the venue's withdrawal policy on behalf of myself and my co-authors.